# Expansion Microscopy Imaging Isotropic Restoration by Unsupervised Deep Learning

**Meng-Yun Wu**[*1]                          WMY4FISH@GMAIL.COM, R11458006@NTU.EDU.TW
**Da-Yu Huang**[*1]                                          R11458005@NTU.EDU.TW
**Ya-Ding Liu**[*2]                                              SF164461@GMAIL.COM
**Li-An Chu**[†2]                                          LACHU@MX.NTHU.EDU.TW
**Gary. Han Chang**[†1]                                  GARYHANCHANG@NTU.EDU.TW

[1] *Institute of Medical Device and Imaging, National Taiwan University College of Medicine, Taipei 100, Taiwan*

[2] *Department of Biomedical Engineering and Environmental Sciences, National Tsing Hua University, Hsinchu 300, Taiwan*

**Editors:** Under Review for MIDL 2023

## Abstract

The development of fluorescence light sheets and expansion microscopy (ExM) in recent years enables the visualization of detailed neural structures to help unlock the secrets of neural functioning. Deep learning techniques have then become essential tools to process the ever-increasing amount of high-quality and high-resolution images. In this study, we developed a single-scale deconvolution model for extracting multiscale deconvoluted response (MDR) from the volumes of microscopy images of neurons and generative models to translate images between the lateral and axial views. The results demonstrated that deep learning as a promising tool in approving image volume quality and comprehension of structural information of light sheet microscopy.

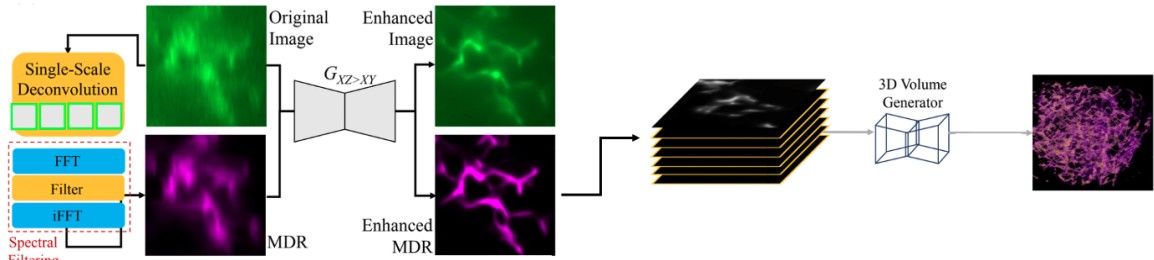

Figure 1: proposed workflow

**Keywords:** domain adaptation, GAN, unsupervised deep learning, expansion microscopy

## 1. Introduction

The development of ExM in recent years has revolutionized the field of biological imaging by allowing visualization of biological structures in sub-millimeter scales. This breakthrough technology has made it possible to observe detailed morphologies of synaptic connections in neurons. However, reconstructing three-dimensional (3D) neuronal morphologies from

---

[*] Contributed equally

[†] Corresponding author

light-sheet microscopy imaging with ExM samples faces two main challenges. One is light-sheet microscopy imaging does not result in a 3D imaging volume with isotropic imaging resolution due to optical sectioning. The other is uneven contrast conditions, due to inhomogeneities in fluorescence labeling and distribution of neurons, often leads to visual ambiguities and poor definition between neuron morphologies and the background. Therefore, there has been tremendous interest in developing deep learning methods to addressing these challenges.

In the study, a single-scale deconvolution model was pre-trained for extracting MDR from the volumes of real microscopy images of neurons. Unsupervised domain-adaptation was subsequently applied by generative adversarial network (GAN) models to translate MDR as well as the original images between the lateral and axial views(Park et al., 2021). This results in an imaging volume with isotropic resolution and imaging quality.

## 2. Methods

### (1) Extract Multi-scale Features with Backbone Deconvolution Model

We constructed a backbone single-scale deconvolution model using idealized neuron structures generated by applying a theoretical PSF, Poisson and Gaussian noise on the synthetic ground truth of neuron structures(Weigert et al., 2018). Compared to the structures of real neurons, the idealized structures lacked the complexities in neuron morphologies and filopodia structures but were easy to be generated in large quantities and able to provide the relationship between the neuron-like structures and the real microscopy images. This supervised model was trained to learn the mapping from synthetic single-scale microscopy images to synthetic ground truth images.

We then approximated the real morphologies of neurons by decomposing the original image into a combination of backbone deconvolution model outputs at 4 physical scales. The multiscale ground-truth of neuron morphologies was subsequently approximated by fusing knowledge at different physical scales using a Fourier Transform. The multiscale deconvolution response (MDR) was defined as the combined features filtered by a learnable filter and converted by an Inverse Fourier Transform. The MDRs at each lateral view slice were subsequently concatenated back to form a 3D volume and resliced on the axial direction to create the MDRs at axial view.

### (2) Isotropic Image Restoration

To create an imaging volume with isotropic image resolution and quality, we enhanced the images as well as MDRs from the axial view by an unsupervised domain adaptation model with CycleGAN(Zhu et al., 2017) architecture. The model contained two generators to translate the data from lateral view to the axial view, and two discriminators to ensure the visual similarity between the enhanced and the referenced image. The original image and the MDRs from either lateral and axial view were concatenated together as the model input and optimized simultaneously to improve their spatial resolution and imaging quality. This enabled our model to utilize the capability of multiscale features to capture neuronal spatial structures across various physical levels.

To create isotropic imaging volumes, the 2D axial slices of the imaging volume were

enhanced independently by the GAN model and concatenated together to from the raw 3D volumes. However, due to the absence of consideration of the adjacent 2D slices during model inference, artifacts due to discontinuities were present in the raw imaging volume as observed from the concatenated lateral view. To overcome this issue, we resliced the original imaging volume along arbitrary directions on the lateral plane and used the resliced slices as input to the model during the inference process. The finalized imaging volume was obtained by averaging all the imaging volumes generated along these different directions.

## 3. Results

As Figure 2 shown, the generated axial view images had their imaging resolution, continuity and quality significantly enhanced to visually resemble the ones from the lateral view. Our model also reduced the influence of imaging artifacts caused by optical slicing on the profiles of neurons. Figure 3 demonstrated the discontinuities between the generated 2D slices were mitigated.

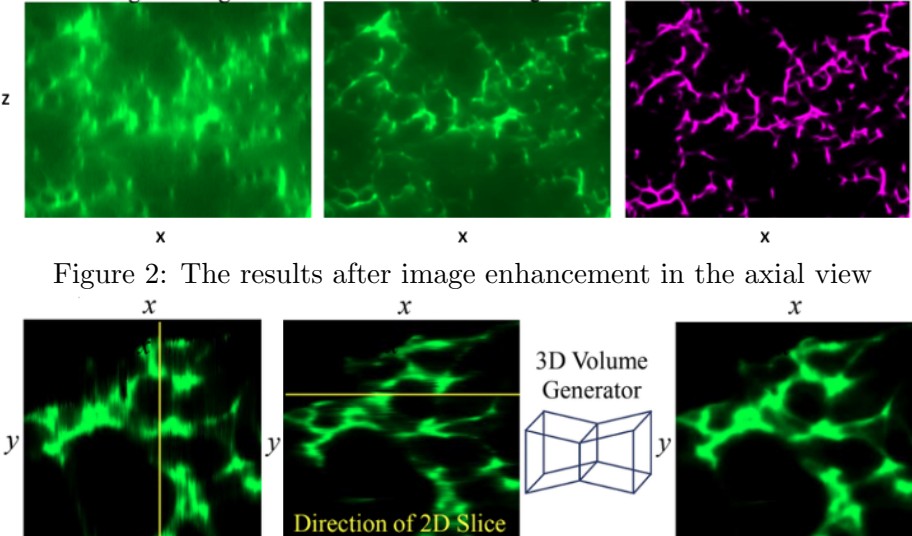

Figure 2: The results after image enhancement in the axial view

Figure 3: L: Discontinuities were present in the raw imaging volume from the lateral view. R: The imaging artifacts were absent from the finalized imaging volume.

## 4. Conclusion

The results of the study show the ability of our proposed model to faithfully enhance the imaging volume of ExM image isotropically, and demonstrated great potential in reconstructing detailed morphologies of neurons with great computational efficiency and required limited amount of manual annotation. These findings are particularly significant in light of the ever-increasing demand for high-resolution imaging techniques that enable researchers to study the complex biological structures of the whole brain and other tissues, and the present model showed broad potential for maximizing the full potential of high-ratio expansion microscopy.

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
