# OpenReview forum: "Expansion Microscopy Imaging Isotropic Restoration by Unsupervised Deep Learning"
_MIDL.io/2023/Short_Paper_Track — MIDL 2023 Short paper track Poster_

### Official Review · Reviewer_Z8h2 · 2023-04-10
**Isotropic restoration of microscopy images**

**Rating:** 8
**Confidence:** 5

**Review:**

This paper enhanced the imaging volume of ExM image isotropically and demonstrated great potential in reconstructing detailed morphologies of neurons
The advantages of the paper include:
+ great computational efficiency and required a limited amount of manual annotation
+ It enables researchers to study the complex biological structures of the whole brain and other tissues
The limitation of the paper includes:
- No quantitative results are provided
- The technical innovation is not clearly conveyed

---

### Official Review · Reviewer_wFzs · 2023-04-21
**Paper 114 review**

**Rating:** 5
**Confidence:** 3

**Review:**

This paper presents a 2-step method to improve image volume quality of light sheet microscopy. First a supervised model is trained as a single-scale deconvolution model using synthetic data of neuron-like structures. Second, a Cycle-GAN model is trained to translate between axial and lateral views. The topic of the paper is an important one and the visual results look promising; however, there are some shortcomings of this paper:
1) The paper is not clear about important details such as loss functions used in training. It is not clear to me what is the loss function that is being used for the CycleGAN that leads to improved resolution
2) There are no quantitative results presented in the paper which reduces the impact of this work. If some quantitative results could be provided even if it is only on synthetic data, this would strengthen the paper.